# OPTIMIZATION AS A MODEL FOR FEW-SHOT LEARNING

**Sachin Ravi**[*] **and Hugo Larochelle**
Twitter, Cambridge, USA
{sachinr,hugo}@twitter.com

## ABSTRACT

Though deep neural networks have shown great success in the large data domain, they generally perform poorly on few-shot learning tasks, where a classifier has to quickly generalize after seeing very few examples from each class. The general belief is that gradient-based optimization in high capacity classifiers requires many iterative steps over many examples to perform well. Here, we propose an LSTM-based *meta-learner* model to learn the exact optimization algorithm used to train another *learner* neural network classifier in the few-shot regime. The parametrization of our model allows it to learn appropriate parameter updates specifically for the scenario where a set amount of updates will be made, while also learning a general initialization of the learner (classifier) network that allows for quick convergence of training. We demonstrate that this meta-learning model is competitive with deep metric-learning techniques for few-shot learning.

## 1 INTRODUCTION

Deep learning has shown great success in a variety of tasks with large amounts of labeled data in image classification (He et al., 2015), machine translation (Wu et al., 2016), and speech modeling (Oord et al., 2016). These achievements have relied on the fact that optimization of these deep, high-capacity models requires many iterative updates across many labeled examples. This type of optimization breaks down in the small data regime where we want to learn from very few labeled examples. In this setting, rather than have one large dataset, we have a set of datasets, each with few annotated examples per class. The motivation for this task lies not only in the fact that humans, even children, can usually generalize after just one example of a given object, but also because models excelling at this task would have many useful applications. Firstly, they would help alleviate data collection as we would not require millions of labeled examples to attain reasonable performance. Furthermore, in many fields, data exhibits the characteristic of having many different classes but few examples per class. Models that are able to generalize from few examples would be able to capture this type of data effectively.

There seem to be two main reasons why gradient-based optimization fails in the face of few labeled examples. Firstly, the variants of gradient-based optimization algorithms, such as momentum (Nesterov, 1983), Adagrad (Duchi et al., 2011), Adadelta (Zeiler, 2012), and ADAM (Kingma & Ba, 2014), weren't designed specifically to perform well under the constraint of a set number of updates. Specifically when applied to non-convex optimization problems, with a reasonable choice of hyperparameters these algorithms don't have very strong guarantees of speed of convergence, beyond that they will eventually converge to a good solution after what could be many millions of iterations. Secondly, for each separate dataset considered, the network would have to start from a random initialization of its parameters, which considerably hurts its ability to converge to a good solution after a few updates. Transfer learning (Caruana, 1995; Bengio et al., 2012; Donahue et al., 2013) can be applied to alleviate this problem by fine-tuning a pre-trained network from another task which has more labelled data; however, it has been observed that the benefit of a pre-trained network greatly decreases as the task the network was trained on diverges from the target task (Yosinski et al., 2014). What is needed is a systematic way to learn a beneficial common initialization that would

---

[*]Work done as an intern at Twitter. Sachin is a PhD student at Princeton University and can be reached at sachinr@princeton.edu.

serve as a good point to start training for the set of datasets being considered. This would provide the same benefits as transfer learning, but with the guarantee that the initialization is an optimal starting point for fine-tuning.

Previous work has suggested one manner in which to acquire quick knowledge from few examples, through the idea of *meta-learning* (Thrun, 1998; Schmidhuber et al., 1997). Meta-learning suggests framing the learning problem at two levels. The first is quick acquisition of knowledge *within* each separate task presented. This process is guided by the second, which involves slower extraction of information learned *across* all the tasks.

We present a method here that addresses the weakness of neutral networks trained with gradient-based optimization on the few-shot learning problem by framing the problem within a meta-learning setting. We propose an LSTM-based *meta-learner* optimizer that is trained to optimize a *learner* neural network classifier. The meta-learner captures both short-term knowledge within a task and long-term knowledge common among all the tasks. By using an objective that directly captures an optimization algorithm's ability to have good generalization performance given only a set number of updates, the meta-learner model is trained to converge a learner classifier to a good solution quickly on each task. Additionally, the formulation of our meta-learner model allows it to learn a task-common initialization for the learner classifier, which captures fundamental knowledge shared among all the tasks.

## 2 TASK DESCRIPTION

We first begin by detailing the meta-learning formulation we use. In the typical machine learning setting, we are interested in a dataset $D$ and usually split $D$ so that we optimize parameters $\theta$ on a training set $D_{train}$ and evaluate its generalization on the test set $D_{test}$. In meta-learning, however, we are dealing with meta-sets $\mathscr{D}$ containing multiple regular datasets, where each $D \in \mathscr{D}$ has a split of $D_{train}$ and $D_{test}$.

We consider the $k$-shot, $N$-class classification task, where for each dataset $D$, the training set consists of $k$ labelled examples for each of $N$ classes, meaning that $D_{train}$ consists of $k \cdot N$ examples, and $D_{test}$ has a set number of examples for evaluation. We note that previous work (Vinyals et al., 2016) has used the term *episode* to describe each dataset consisting of a training and test set.

In meta-learning, we thus have different meta-sets for meta-training, meta-validation, and meta-testing ($\mathscr{D}_{meta-train}$, $\mathscr{D}_{meta-validation}$, and $\mathscr{D}_{meta-test}$, respectively). On $\mathscr{D}_{meta-train}$, we are interested in training *a learning procedure* (the meta-learner) that can take as input one of its training sets $D_{train}$ and produce a classifier (the learner) that achieves high average classification performance on its corresponding test set $D_{test}$. Using $\mathscr{D}_{meta-validation}$ we can perform hyper-parameter selection of the meta-learner and evaluate its generalization performance on $\mathscr{D}_{meta-test}$.

For this formulation to correspond to the few-shot learning setting, each training set in datasets $D \in \mathscr{D}$ will contain few labeled examples (we consider $k = 1$ or $k = 5$), that must be used to generalize to good performance on the corresponding test set. An example of this formulation is given in Figure 1.

## 3 MODEL

We now move to the description of our proposed model for meta-learning.

### 3.1 MODEL DESCRIPTION

Consider a single dataset, or episode, $D \in \mathscr{D}_{meta-train}$. Suppose we have a *learner* neural net classifier with parameters $\theta$ that we want to train on $D_{train}$. The standard optimization algorithms used to train deep neural networks are some variant of gradient descent, which uses updates of the form

$$\theta_t = \theta_{t-1} - \alpha_t \nabla_{\theta_{t-1}} \mathcal{L}_t, \tag{1}$$

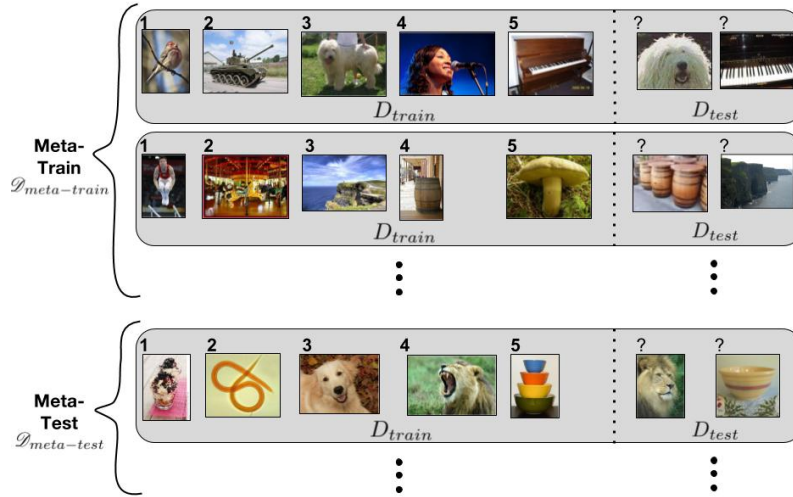

Figure 1: Example of meta-learning setup. The top represents the meta-training set $\mathscr{D}_{meta-train}$, where inside each gray box is a separate dataset that consists of the training set $D_{train}$ (left side of dashed line) and the test set $D_{test}$ (right side of dashed line). In this illustration, we are considering the 1-shot, 5-class classification task where for each dataset, we have one example from each of 5 classes (each given a label 1-5) in the training set and 2 examples for evaluation in the test set. The meta-test set $\mathscr{D}_{meta-test}$ is defined in the same way, but with a different set of datasets that cover classes not present in any of the datasets in $\mathscr{D}_{meta-train}$ (similarly, we additionally have a meta-validation set that is used to determine hyper-parameters).

where $\theta_{t-1}$ are the parameters of the learner after $t-1$ updates, $\alpha_t$ is the learning rate at time $t$, $\mathcal{L}_t$ is the loss optimized by the learner for its $t^{\text{th}}$ update, $\nabla_{\theta_{t-1}}\mathcal{L}_t$ is the gradient of that loss with respect to parameters $\theta_{t-1}$, and $\theta_t$ is the updated parameters of the learner.

Our key observation that we leverage here is that this update resembles the update for the cell state in an LSTM (Hochreiter & Schmidhuber, 1997)

$$c_t = f_t \odot c_{t-1} + i_t \odot \tilde{c}_t, \tag{2}$$

if $f_t = 1, c_{t-1} = \theta_{t-1}, i_t = \alpha_t$, and $\tilde{c}_t = -\nabla_{\theta_{t-1}}\mathcal{L}_t$.

Thus, we propose training a *meta-learner* LSTM to learn an update rule for training a neural network. We set the cell state of the LSTM to be the parameters of the learner, or $c_t = \theta_t$, and the candidate cell state $\tilde{c}_t = \nabla_{\theta_{t-1}}\mathcal{L}_t$, given how valuable information about the gradient is for optimization. We define parametric forms for $i_t$ and $f_t$ so that the meta-learner can determine optimal values through the course of the updates.

Let us start with $i_t$, which corresponds to the learning rate for the updates. We let

$$i_t = \sigma\left(\mathbf{W}_I \cdot \left[\nabla_{\theta_{t-1}}\mathcal{L}_t, \mathcal{L}_t, \theta_{t-1}, i_{t-1}\right] + \mathbf{b}_I\right),$$

meaning that the learning rate is a function of the current parameter value $\theta_{t-1}$, the current gradient $\nabla_{\theta_{t-1}}\mathcal{L}_t$, the current loss $\mathcal{L}_t$, and the previous learning rate $i_{t-1}$. With this information, the meta-learner should be able to finely control the learning rate so as to train the learner quickly while avoiding divergence.

As for $f_t$, it seems possible that the optimal choice isn't the constant 1. Intuitively, what would justify shrinking the parameters of the learner and forgetting part of its previous value would be if the learner is currently in a bad local optima and needs a large change to escape. This would correspond to a situation where the loss is high but the gradient is close to zero. Thus, one proposal for the forget gate is to have it be a function of that information, as well as the previous value of the forget gate:

$$f_t = \sigma\left(\mathbf{W}_F \cdot \left[\nabla_{\theta_{t-1}}\mathcal{L}_t, \mathcal{L}_t, \theta_{t-1}, f_{t-1}\right] + \mathbf{b}_F\right).$$

Additionally, notice that we can also learn the initial value of the cell state $c_0$ for the LSTM, treating it as a parameter of the meta-learner. This corresponds to the initial weights of the classifier (that

the meta-learner is training). Learning this initial value lets the meta-learner determine the optimal initial weights of the learner so that training begins from a beneficial starting point that allows optimization to proceed rapidly. Lastly, note that though the meta-learner's update rule matches the cell state update of the LSTM, the meta-learner also bears similarity to the GRU (Cho et al., 2014) hidden state update, with the exception that the forget and input gates aren't tied to sum to one.

## 3.2 PARAMETER SHARING & PREPROCESSING

Because we want our meta-learner to produce updates for deep neural networks, which consist of tens of thousands of parameters, to prevent an explosion of meta-learner parameters we need to employ some sort of parameter sharing. Thus as in Andrychowicz et al. (2016), we share parameters across the coordinates of the learner gradient. This means each coordinate has its own hidden and cell state values but the LSTM parameters are the same across all coordinates. This allows us to use a compact LSTM model and additionally has the nice property that the same update rule is used for each coordinate, but one that is dependent on the respective history of each coordinate during optimization. We can easily implement parameter sharing by having the input be a batch of gradient coordinates and loss inputs $(\nabla_{\theta_{t,i}} \mathcal{L}_t, \mathcal{L}_t)$ for each dimension $i$.

Because the different coordinates of the gradients and the losses can be of very different magnitudes, we need to be careful in normalizing the values so that the meta-learner is able to use them properly during training. Thus, we also found that the preprocessing method of Andrychowicz et al. (2016) worked well when applied to both the dimensions of the gradients and the losses at each time step:

$$x \rightarrow \begin{cases} \left( \frac{\log(|x|)}{p}, \operatorname{sgn}(x) \right) & \text{if } |x| \geq e^{-p} \\ (-1, e^p x) & \text{otherwise} \end{cases}$$

This preprocessing adjusts the scaling of gradients and losses, while also separating the information about their magnitude and their sign (the latter being mostly useful for gradients). We found that the suggested value of $p = 10$ in the above formula worked well in our experiments.

## 3.3 TRAINING

The question now is how do we train the LSTM meta-learner model to be effective at few-shot learning tasks? As observed in Vinyals et al. (2016), in order to perform well at this task, it is key to have training conditions match those of test time. During evaluation of the meta-learning, for each dataset (episode), $D = (D_{train}, D_{test}) \in \mathscr{D}_{meta-test}$, a good meta-learner model will, given a series of learner gradients and losses on the training set $D_{train}$, suggest a series of updates for the classifier that pushes it towards good performance on the test set $D_{test}$.

Thus to match test time conditions, when considering each dataset $D \in \mathscr{D}_{meta-train}$, the training objective we use is the loss $\mathcal{L}_{test}$ of the produced classifier on $D$'s test set $D_{test}$. While iterating over the examples in $D$'s training set $D_{train}$, at each time step $t$ the LSTM meta-learner receives $(\nabla_{\theta_{t-1}} \mathcal{L}_t, \mathcal{L}_t)$ from the learner (the classifier) and proposes the new set of parameters $\theta_t$. The process repeats for $T$ steps, after which the classifier and its final parameters are evaluated on the test set to produce the loss that is then used to train the meta-learner. The training algorithm is described in Algorithm 1 and the corresponding computational graph is shown in Figure 2.

### 3.3.1 GRADIENT INDEPENDENCE ASSUMPTION

Notice that our formulation would imply that the losses $\mathcal{L}_t$ and gradients $\nabla_{\theta_{t-1}} \mathcal{L}_t$ of the learner are dependent on the parameters of the meta-learner. Gradients on the meta-learner's parameters should normally take this dependency into account. However, as discussed by Andrychowicz et al. (2016), this complicates the computation of the meta-learner's gradients. Thus, following Andrychowicz et al. (2016), we make the simplifying assumption that these contributions to the gradients aren't important and can be ignored, which allows us to avoid taking second derivatives, a considerably expensive operation. We were still able to train the meta-learner effectively in spite of this simplifying assumption.

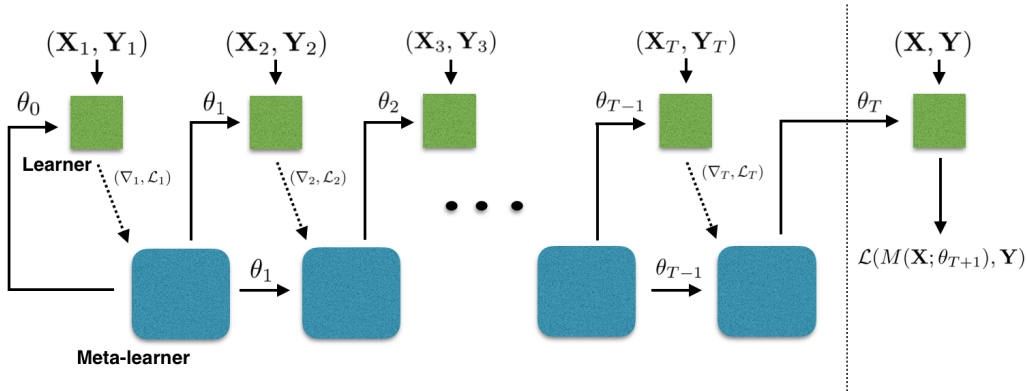

Figure 2: Computational graph for the forward pass of the meta-learner. The dashed line divides examples from the training set $D_{train}$ and test set $D_{test}$. Each $(\mathbf{X}_i, \mathbf{Y}_i)$ is the $i^{th}$ batch from the training set whereas $(\mathbf{X}, \mathbf{Y})$ is all the elements from the test set. The dashed arrows indicate that we do not back-propagate through that step when training the meta-learner. We refer to the learner as $M$, where $M(\mathbf{X}; \theta)$ is the output of learner $M$ using parameters $\theta$ for inputs $\mathbf{X}$. We also use $\nabla_t$ as a shorthand for $\nabla_{\theta_{t-1}} \mathcal{L}_t$.

### 3.3.2 INITIALIZATION OF META-LEARNER LSTM

When training LSTMs, it is advised to initialize the LSTM with small random weights and to set the forget gate bias to a large value so that the forget gate is initialized to be close to 1, thus enabling gradient flow (Zaremba, 2015). In addition to the forget gate bias setting, we found that we needed to initialize the input gate bias to be small so that the input gate value (and thus the learning rate) used by the meta-learner LSTM starts out being small. With this combined initialization, the meta-learner starts close to normal gradient descent with a small learning rate, which helps initial stability of training.

### 3.4 BATCH NORMALIZATION

Batch Normalization (Ioffe & Szegedy, 2015) is a recently proposed method to stabilize and thus speed up learning of deep neural networks by reducing internal covariate shift within the learner's hidden layers. This reduction is achieved by normalizing each layer's pre-activation, by subtracting by the mean and dividing by the standard deviation. During training, the mean and standard deviation are estimated using the current batch being trained on, whereas during evaluation a running average of both statistics calculated on the training set is used. We need to be careful with batch normalization for the learner network in the meta-learning setting, because we do not want to collect mean and standard deviation statistics during meta-testing in a way that allows information to leak between different datasets (episodes), being considered. One easy way to prevent this issue is to not collect statistics at all during the meta-testing phase, but just use our running averages from meta-training. This, however, has a bad impact on performance, because we have changed meta-training and meta-testing conditions, causing the meta-learner to learn a method of optimization that relies on batch statistics which it now does not have at meta-testing time. In order to keep the two phases as similar as possible, we found that a better strategy was to collect statistics for each dataset $D \in \mathscr{D}$ during $\mathscr{D}_{meta-test}$, but then erase the running statistics when we consider the next dataset. Thus, during meta-training, we use batch statistics for both the training and testing set whereas during meta-testing, we use batch statistics for the training set (and to compute our running averages) but then use the running averages during testing. This does not cause any information to leak between different datasets, but also allows the meta-learner to be trained on conditions that are matched between training and testing. Lastly, because we are doing very few training steps, we computed the running averages so that higher preference is given to the later values.

---

**Algorithm 1** Train Meta-Learner

---

**Input**: Meta-training set $\mathscr{D}_{meta-train}$, Learner $M$ with parameters $\theta$, Meta-Learner $R$ with parameters $\Theta$.

---

1: $\Theta_0 \leftarrow$ random initialization
2:
3: **for** $d = 1, n$ **do**
4: $D_{train}, D_{test} \leftarrow$ random dataset from $\mathscr{D}_{meta-train}$
5: $\theta_0 \leftarrow c_0$ ▷ Intialize learner parameters
6:
7: **for** $t = 1, T$ **do**
8: $\mathbf{X}_t, \mathbf{Y}_t \leftarrow$ random batch from $D_{train}$
9: $\mathcal{L}_t \leftarrow \mathcal{L}(M(\mathbf{X}_t; \theta_{t-1}), \mathbf{Y}_t)$ ▷ Get loss of learner on train batch
10: $c_t \leftarrow R((\nabla_{\theta_{t-1}}\mathcal{L}_t, \mathcal{L}_t); \Theta_{d-1})$ ▷ Get output of meta-learner using Equation 2
11: $\theta_t \leftarrow c_t$ ▷ Update learner parameters
12: **end for**
13:
14: $\mathbf{X}, \mathbf{Y} \leftarrow D_{test}$
15: $\mathcal{L}_{test} \leftarrow \mathcal{L}(M(\mathbf{X}; \theta_T), \mathbf{Y})$ ▷ Get loss of learner on test batch
16: Update $\Theta_d$ using $\nabla_{\Theta_{d-1}}\mathcal{L}_{test}$ ▷ Update meta-learner parameters
17:
18: **end for**

---

## 4 RELATED WORK

While this work falls within the broad literature of transfer learning in general, we focus here on positioning it relative to previous work on meta-learning and few-shot learning.

### 4.1 META-LEARNING

Meta-learning has a long history, but has grown to prominence recently as many have advocated for it as a key to achieving human-level intelligence in the future (Lake et al., 2016). The ability to learn at two levels (learning within each task presented, while accumulating knowledge about the similarities and differences between tasks) is seen as being crucial to improving AI. Previous work has used a variety of techniques in the meta-learning setting.

Schmidhuber (1992; 1993) explored using networks that learn how to modify their own weights over a number of computations steps on the input. The updating of the weights is defined in a parametric form that allows the prediction and weight-change process to be differentiable end-to-end. The work of Bengio et al. (1990; 1995) and Bengio (1993) considered learning update rules for neural networks that are biologically plausible. This property is enforced by allowing the parametric form of the update to only have as input local information at each hidden unit to determine the weight change. Different optimization methods, such as genetic programming or simulated annealing, are used to train the learning rule.

In Santoro et al. (2016), a memory-augmented neural network is trained to learn how to store and retrieve memories to use for each classification task. The work of Andrychowicz et al. (2016) uses an LSTM to train a neural network; however, they are interested in learning a general optimization algorithm to train neural networks for large-scale classification, whereas we are interested in the few-shot learning problem. This work also builds upon Hochreiter et al. (2001) and Bosc, both of which used LSTMs to train multi-layer perceptrons to learn on binary classification and time-series prediction tasks. Another related method is the work of Bertinetto et al. (2016), who train a meta-learner to map a training example to the weights of a neural network that is then used to classify future examples from this class; however, unlike our method the classifier network is directly produced rather than being fine-tuned after multiple training steps. Our work also bears similarity to Maclaurin et al. (2015), who tune the hyperparameters of gradient descent with momentum by backpropagating through the chain of gradient steps to optimize the validation performance.

## 4.2 Few-shot learning

The best performing methods for few-shot learning have been mainly metric learning methods. Deep siamese networks (Koch, 2015) train a convolutional network to embed examples so that items in the same class are close while items in different classes are far away, according to some distance metric. Matching networks (Vinyals et al., 2016) refine this idea so that training and testing conditions match, by defining a differentiable nearest neighbor loss involving the cosine similarities of embeddings produced by a convolutional network.

## 5 Evaluation

In this section, we describe the results of experiments, examining the properties of our model and comparing our method's performance against different approaches[*]. Following Vinyals et al. (2016), we consider the $k$-shot, $N$-class classification setting where a meta-learner trains on many related but small training sets of $k$ examples for each of $N$ classes. We first split the list of all classes in the data into disjoint sets and assign them to each meta-set of meta-training, meta-validation, and meta-testing. To generate each instance of a $k$-shot, $N$-class task dataset $D = (D_{train}, D_{test}) \in \mathscr{D}$, we do the following: we first sample $N$ classes from the list of classes corresponding to the meta-set we consider. We then sample $k$ examples from each of those classes. These $k$ examples together compose the training set $D_{train}$. Then, an additional fixed amount of the rest of the examples are sampled to yield a test set $D_{test}$. We generally have 15 examples per class in the test sets. When training the meta-learner, we iterate by sampling these datasets (episodes) repeatedly. For meta-validation and meta-testing, however, we produce a fixed number of these datasets to evaluate each method. We produce enough datasets to ensure that the confidence interval of the mean accuracy is small.

For the learner, we use a simple CNN containing 4 convolutional layers, each of which is a $3 \times 3$ convolution with 32 filters, followed by batch normalization, a ReLU non-linearity, and lastly a $2 \times 2$ max-pooling. The network then has a final linear layer followed by a softmax for the number of classes being considered. The loss function $\mathcal{L}$ is the average negative log-probability assigned by the learner to the correct class. For the meta-learner, we use a 2-layer LSTM, where the first layer is a normal LSTM and the second layer is our modified LSTM meta-learner. The gradients and losses are preprocessed and fed into the first layer LSTM, and the regular gradient coordinates are also used by the second layer LSTM to implement the state update rule shown in (1). At each time step, the learner's loss and gradient is computed on a batch consisting of the entire training set $D_{train}$, because we consider training sets with only a total of 5 or 25 examples. We train our LSTM with ADAM using a learning rate of 0.001 and with gradient clipping using a value of 0.25.

## 5.1 Experiment Results

The Mini-ImageNet dataset was proposed by Vinyals et al. (2016) as a benchmark offering the challenges of the complexity of ImageNet images, without requiring the resources and infrastructure necessary to run on the full ImageNet dataset. Because the exact splits used in Vinyals et al. (2016) were not released, we create our own version of the Mini-Imagenet dataset by selecting a random 100 classes from ImageNet and picking 600 examples of each class. We use 64, 16, and 20 classes for training, validation and testing, respectively. We consider 1-shot and 5-shot classification for 5 classes. We use 15 examples per class for evaluation in each test set. We compare against two baselines and a recent metric-learning technique, Matching Networks (Vinyals et al., 2016), which has achieved state-of-the-art results in few-shot learning. The results are shown in Table 1.

The first baseline we use is a nearest-neighbor baseline (*Baseline-nearest-neighbor*), where we first train a network to classify between all the classes jointly in the original meta-training set. At meta-test time, for each dataset $D$, we embed all the items in the training set using our trained network and then use nearest-neighbor matching among the embedded training examples to classify each test example. The second baseline we use (*Baseline-finetune*) represents a coarser version of our meta-learner model. As in the first baseline, we start by training a network to classify jointly between all classes in the meta-training set. We then use the meta-validation set to search over SGD hyperparameters, where each training set is used to fine-tune the pre-trained network before evaluating on

---

[*]Code can be found at `https://github.com/twitter/meta-learning-lstm`.

| Model | 5-class | |
|---|---|---|
| | 1-shot | 5-shot |
| Baseline-finetune | $28.86 \pm 0.54\%$ | $49.79 \pm 0.79\%$ |
| Baseline-nearest-neighbor | $41.08 \pm 0.70\%$ | $51.04 \pm 0.65\%$ |
| Matching Network | $\mathbf{43.40 \pm 0.78\%}$ | $51.09 \pm 0.71\%$ |
| Matching Network FCE | $\mathbf{43.56 \pm 0.84\%}$ | $55.31 \pm 0.73\%$ |
| Meta-Learner LSTM (OURS) | $\mathbf{43.44 \pm 0.77\%}$ | $\mathbf{60.60 \pm 0.71\%}$ |

Table 1: Average classification accuracies on Mini-ImageNet with 95% confidence intervals. Marked in bold are the best results for each scenario, as well as other results with an overlapping confidence interval.

the test set. We use a fixed number of updates for fine tuning and search over the learning rate and learning rate decay used during the course of these updates.

For Matching Networks, we implemented our own version of both the basic and the fully-conditional embedding (FCE) versions. In the basic version, a convolutional network is trained to learn independent embeddings for examples in the training and test set. In the FCE version, a bidirectional-LSTM is used to learn an embedding for the training set such that each training example's embedding is also a function of all the other training examples. Additionally, an attention-LSTM is used so that a test example embedding is also a function of all the embeddings of the training set. We do not consider fine-tuning the network using the train set during meta-testing to improve performance as mentioned in Vinyals et al. (2016), but do note that our meta-learner could also be fine-tuned using this data. Note that to remain consistent with Vinyals et al. (2016), our baseline and matching net convolutional networks have 4 layers each with 64 filters. We also added dropout to each convolutional block in matching nets to prevent overfitting.

For our meta-learner, we train different models for the 1-shot and 5-shot tasks, that make 12 and 5 updates, respectively. We noticed that better performance for each task was attained if the meta-learner is explicitly trained to do the set number of updates during meta-training that will be used during meta-testing.

We attain results that are much better than the baselines discussed and competitive with Matching Networks. For 5-shot, we are able to do much better than Matching Networks, whereas for 1-shot, the confidence interval for our performance intersects the interval for Matching Networks. Again, we note that the numbers do not match the ones provided by Vinyals et al. (2016) simply because we created our version of the dataset and implemented our own versions of their model. It is interesting to note that the fine-tuned baseline is worse than the nearest-neighbor baseline. Because we are not regularizing the classifier, with very few updates the fine-tuning model overfits, especially in the 1-shot case. This propensity to overfit speaks to the benefit of meta-training the initialization of the classifier end-to-end as is done in the meta-learning LSTM.

## 5.2 VISUALIZATION OF META-LEARNER

We also visualize the optimization strategy learned by the meta-learner, in Figure 3. We can look at the $i_t$ and $f_t$ gate values in Equation 2 at each update step, to try to get an understanding of how the meta-learner updates the learner during training. We visualize the gate values while training on different datasets $D_{train}$, to observe whether there are variations between training sets. We consider both 1-shot and 5-shot classification settings, where the meta-learner is making 10 and 5 updates, respectively. For the forget gate values for both tasks, the meta-learner seems to adopt a simple weight decay strategy that seems consistent across different layers. The input gate values are harder to interpret to glean the meta-learner's strategy. However, there seems to a be a lot of variability between different datasets, indicating that the meta-learner isn't simply learning a fixed optimization strategy. Additionally, there seem to be differences between the two tasks, suggesting that the meta-learner has adopted different methods to deal with the different conditions of each setting.

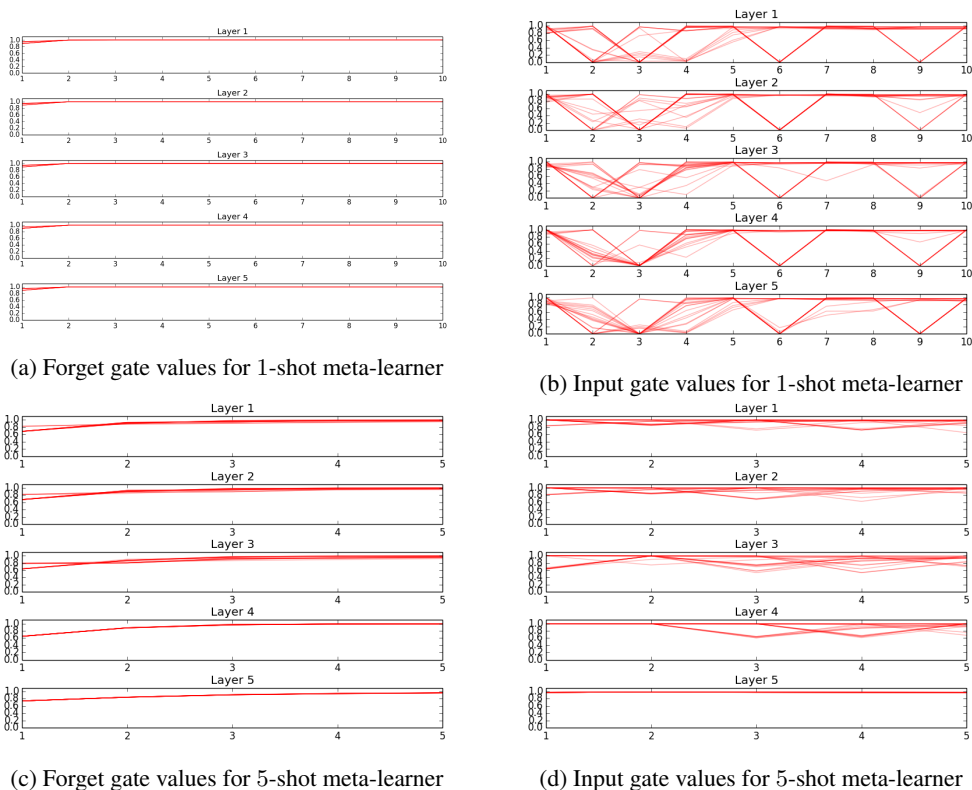

(a) Forget gate values for 1-shot meta-learner

(b) Input gate values for 1-shot meta-learner

(c) Forget gate values for 5-shot meta-learner

(d) Input gate values for 5-shot meta-learner

Figure 3: Visualization of the input and forget values output by the meta-learner during the course of its updates. Layers $1 - 4$ represent the values for a randomly selected parameter from the $4$ convolutional layers and layer $5$ represents the values for a random parameter from fully-connected layer. The different curves represent training steps on different datasets.

## 6 CONCLUSION

We described an LSTM-based model for meta-learning, which is inspired from the parameter updates suggested by gradient descent optimization algorithms. Our LSTM meta-learner uses its state to represent the learning updates of the parameters of a classifier. It is trained to discover both a good initialization for the learner's parameters, as well as a successful mechanism for updating the learner's parameters to a given small training set for some new classification task. Our experiments demonstrate that our approach outperforms natural baselines and is competitive to the state-of-the-art in metric learning for few-shot learning.

In this work, we focused our study to the few-shot and few-classes setting. However, it would be more valuable to train meta-learners that can perform well across a full spectrum of settings, i.e. for few or lots of training examples and for few or lots of possible classes. Our future work will thus consider moving towards this more challenging scenario.

ACKNOWLEDGMENTS

We thank Jake Snell, Kevin Swersky, and Oriol Vinyals for helpful discussions of this work.

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
