# Peer review of "Optimization as a Model for Few-Shot Learning"

_ICLR 2017 — accepted_

[Author Response · Sachin Ravi · 02 Dec 2016]
**Typo for equation 2 fixed**

In the 3rd revision, we fixed a minor typo in description of equation 2.

[Public Comment · (anonymous) · 09 Dec 2016]
**nice paper**

Nice paper! I have one orthogonal question for the mini-imageNet dataset:

"we create our own version of the Mini-Imagenet dataset by selecting a random 100 classes from ImageNet and picking 600 examples of each class". 

I am curious how sensitive with random selected classes and examples? It would be great if the exact split or the whole dataset is shared publicly, so others can repeat experiments and make comparison on a fixed benchmark dataset.

[Official Review · AnonReviewer1 · rating 8 · confidence 4 · 18 Dec 2016 (modified: 21 Jan 2017)]
**Strong paper but presentation unclear at times**

In light of the authors' responsiveness and the updates to the manuscript -- in particular to clarify the meta-learning task -- I am updating my score to an 8.

-----

This manuscript proposes to tackle few-shot learning with neural networks by leveraging meta-learning, a classic idea that has seen a renaissance in the last 12 months. The authors formulate few-shot learning as a sequential meta-learning problem: each "example" includes a sequence of batches of "training" pairs, followed by a final "test" batch. The inputs at each "step" include the outputs of a "base learner" (e.g., training loss and gradients), as well as the base learner's current state (parameters). The paper applies an LSTM to this meta-learning problem, using the inner memory cells in the *second* layer to directly model the updated parameters of the base learner. In doing this, they note similarities between the respective update rules of LSTM memory cells and gradient descent. Updates to the LSTM meta-learner are computed based on the base learner's prediction loss for the final "test" batch. The authors make several simplifying assumptions, such as sharing weights across all second layer cells (analogous to using the same learning rate for all parameters). The paper recreates the Mini-ImageNet data set proposed in Vinyals et al 2016, and shows that the meta-learner LSTM is competitive with the current state-of-the-art (Matchin Networks, Vinyals 2016) on 1- and 5-shot learning.

Strengths:
- It is intriguing -- and in hindsight, natural -- to cast the few-shot learning problem as a sequential (meta-)learning problem. While the authors did not originate the general idea of persisting learning across a series of learning problems, I think it is fair to say that they have advanced the state of the art, though I cannot confidently assert its novelty as I am not deeply familiar with recent work on meta-learning.
- The proposed approach is competitive with and outperforms Vinyals 2016 in 1-shot and 5-shot Mini-ImageNet experiments.
- The base learner in this setting (simple ConvNet classifier) is quite different from the nearest-neighbor-on-top-of-learned-embedding approach used in Vinyals 2016. It is always exciting when state-of-the-art results can be reported using very different approaches, rather than incremental follow-up work.
- As far as I know, the insight about the relationship between the memory cell and gradient descent updates is novel here. It is interesting regardless.
- The paper offers several practical insights about how to design and train an LSTM meta-learner, which should make it easier for others to replicate this work and apply these ideas to new problems. These include proper initialization, weight sharing across coordinates, and the importance of normalizing/rescaling the loss, gradient, and parameter inputs. Some of the insights have been previously described (the importance of simulating test conditions during meta-training; assuming independence between meta-learner and base learner parameters when taking gradients with respect to the meta-learner parameters), but the discussion here is useful nonetheless.

Weaknesses:
- The writing is at times quite opaque. While it describes very interesting work, I would not call the paper an enjoyable read. It took me multiple passes (as well as consulting related work) to understand the general learning problem. The task description in Section 2 (Page 2) is very abstract and uses notation and language that is not common outside of this sub-area. The paper could benefit from a brief concrete example (based on MNIST is fine), perhaps paired with a diagram illustrating a sequence of few-shot learning tasks. This would definitely make it accessible to a wider audience.
- Following up on that note, the precise nature of the N-class, few-shot learning problem here is unclear to me. Specifically, the Mini-ImageNet data set has 100 labels, of which 64/16/20 are used during meta-training/validation/testing. Does this mean that only 64/100 classes are observed through meta-training? Or does it mean that only 64/100 are observed in each batch, but on average all 100 are observed during meta-training? If it's the former, how many outputs does the softmax layer of the ConvNet base learner have during meta-training? 64 (only those observed in training) or 100 (of which 36 are never observed)? Many other details like these are unclear (see question).
- The plots in Figure 2 are pretty uninformative in and of themselves, and the discussion section offers very little insight around them.

This is an interesting paper with convincing results. It seems like a fairly clear accept, but the presentation of the ideas and work therein could be improved. I will definitely raise my score if the writing is improved.

[Official Review · AnonReviewer2 · rating 6 · confidence 4 · 20 Dec 2016]
**An interesting work to understand gradient descent as recurrent process**

This paper describes a new approach to meta learning by interpreting the SGD update rule as gated recurrent model with trainable parameters. The idea is original and important for research related to transfer learning. The paper has a clear structure, but clarity could be improved at some points.

Pros:

- An interesting and feasible approach to meta-learning
- Competitive results and proper comparison to state-of-the-art
- Good recommendations for practical systems

Cons:

- The analogy would be closer to GRUs than LSTMs
- The description of the data separation in meta sets is hard to follow and could be visualized
- The experimental evaluation is only partly satisfying, especially the effect of the parameters of i_t and f_t would be of interest
- Fig 2 doesn't have much value

Remarks:

- Small typo in 3.2: "This means each coordinate has it" -> its

> We plan on releasing the code used in our evaluation experiments.

This would certainly be a major plus.

[Official Review · AnonReviewer3 · rating 9 · confidence 5 · 21 Dec 2016]
**nice paper**

This work presents an LSTM based meta-learning framework to learn the optimization algorithm of a another learning algorithm (here a NN).
The paper is globally well written and the presentation of the main material is clear. The crux of the paper: drawing the parallel between Robbins Monroe update rule and the LSTM update rule and exploit it to satisfy the two main desiderata of few shot learning (1- quick acquisition of new knowledge, 2- slower extraction of general transferable knowledge) is intriguing. 

Several tricks re-used from (Andrychowicz et al. 2016)  such as parameter sharing and normalization, and novel design choices (specific implementation of batch normalization) are well  motivated. 
The experiments are convincing. This is a strong paper. My only concerns/questions are the following:

1. Can it be redundant to use the loss, gradient and parameters as input to the meta-learner? Did you do ablative studies to make sure simpler combinations are not enough.
2. It would be great if other architectural components of the network can be learned in a similar fashion (number of neurons, type of units, etc.). Do you have an opinion about this?
3. The related work section (mainly focused on meta learning) is a bit shallow. Meta-learning is a rather old topic and similar approaches have been tried to solve the same problem even if they were not using LSTMs:
     - Samy Bengio PhD thesis (1989) is all about this ;-)
     - Use of genetic programming for the search of a new learning rule for neural networks (S. Bengio, Y. Bengio, and J. Cloutier. 1994)
     - I am convince Schmidhuber has done something, make sure you find it and update related work section.  

Overall, I like the paper. I believe the discussed material is relevant to a wide audience at ICLR.

[Author Response · Sachin Ravi · 08 Jan 2017]
**Revision in response to reviews uploaded**

We have uploaded a new revision with changes suggested by reviews:

1. Added figure with concrete example of meta-learning framework
2. Reworded some text in order to improve explanation of meta-learning
3. Added citations suggested by reviewers

[Final Decision · Program Chairs · 06 Feb 2017]
**ICLR committee final decision**

The authors propose a meta-learner to address the problem of few-shot learning. The algorithm is interesting, and results are convincing. It's a very timely paper that will receive attention in the community. All three reviewers recommend an accept, with two being particularly enthusiastic. The authors also addressed some issues raised by the more negative reviewer. The AC also agrees that the writing needs a little more work to improve clarity. Overall, this is a clear accept.